# Pediatric Microlaryngoscopy Experiences in a Tertiary Hospital: A Retrospective Analyses of 105 Procedures

**DOI:** 10.3390/medicina60111729

**Published:** 2024-10-22

**Authors:** Elvan Ocmen, Hale Aksu Erdost, Sezin Ersoy, Idris Akdeniz, Taner Erdag

**Affiliations:** 1Department of Anesthesiology and Reanimation, Dokuz Eylul University, 35220 İzmir, Türkiye; drhaleaksuerdost@gmail.com (H.A.E.); sezin.ersoy@deu.edu.tr (S.E.); 2Department of Ear, Nose and Throat Surgery, Dokuz Eylul University, 35220 İzmir, Türkiye; idris.akdeniz@deu.edu.tr (I.A.); taner.erdag@deu.edu.tr (T.E.)

**Keywords:** pediatric anesthesia, airway management, laryngoscopy

## Abstract

*Background and Objectives*: Anesthesia for pediatric microlaryngoscopy/direct laryngoscopy and bronchoscopy (DLB) is very challenging. Airway management methods can vary from tubeless techniques to endotracheal intubation. In this study, we analyzed the pediatric DLB patients operated on in our tertiary hospital. *Materials and Methods*: After the ethics committee’s approval, we retrospectively searched the pediatric DLB patients operated on between 2018 and 2023. Demographic data, airway management, pathology, operation time, and complications were analyzed. *Results*: Fifty-seven pediatric patients and 105 procedures were analyzed. The most common pathology was subglottic stenosis (29.1%). More than half of the patients were younger than one year old (56.1%). The airway management was performed with intermittent mask ventilation (37.2%), endotracheal tube (33.3%), and tracheostomy cannula (29.5%). Intermittent mask ventilation was the airway management type in 66.0% of the infants. *Conclusions*: Here, we report our pediatric DLB experiences. Airway management is challenging and is dependent on the age and pathology of the child, and planned surgery. Excessive attention is required during airway surgeries such as DLB.

## 1. Introduction

Airway surgery can be performed at every age group but pediatric airway surgery is more challenging than adults. Especially laryngeal surgery is much harder for young age groups. Anesthesiologists should provide a good surgical view while maintaining adequate ventilation, oxygenation, and anesthesia. In small age groups, since the endotracheal tube blocks the view of the surgeon during laryngeal surgery, tubeless techniques should be used. These are apneic anesthesia with mask intermittent ventilation, jet-ventilation or spontaneous ventilation, or apnea with high-flow nasal cannula. Each technique has its own benefits and hazards [1,2].

Microlaryngoscopy/direct laryngoscopy and bronchoscopy (DLB) is one of the most common procedures of otolaryngology. The common pathologies for DLB in the pediatric age group are; laryngomalacia, laryngeal clefts, recurrent respiratory papillomatosis, subglottic cysts, subglottic hemangioma, and subglottic stenosis. Airway management of a child with such airway pathologies is challenging and they are more prone to complications related to airway management at every step. Some operations are planned only for exposure of the airway but some surgeries are planned for interventions. The plan of the surgery changes the operation time, the type of airway management, and the risk of complications [2].

In our hospital, we use both the tubeless technique and endotracheal intubation during DLB according to the child’s age and operation. In this observational study, we aimed to investigate the pediatric DLB patients performed in our ear, nose, and throat (ENT) operation room for airway management type and complications.

## 2. Materials and Methods

After the approval of the Dokuz Eylul University Observational Studies ethics committee (2023/31-20), we searched the pediatric (younger than 18 years old) DLB patients operated on in our ENT operation room between 2018 and 2023. Demographic data of the patients, diagnosis, performed operation, operation time, airway management type, and complications were recorded.

Patients older than 18 years old, the ones whose records could not be accessed, and those on whom DLB was not performed were excluded from the study. For the patients that had recurrent DLB, the DLBs performed after the patient exceeded 18 years of age were also excluded.

The known pathologies that are the justification for DLB or the diagnosis detected during the DLB were recorded. The DLBs in which any larynx pathology was not detected were noted as diagnostic DLBs.

If the endoscopic record of the child was accessed by the ENT surgeon, the pediatric anesthesiologist reviewed the record before the day of surgery.

Intravenous (IV) induction was performed with atropine, propofol, and remifentanil when an IV cannula was present. For children without an IV cannula, after inhalational induction, IV access was established. Anesthesia was maintained with propofol and remifentanil infusions to ensure the depth of anesthesia during apneic or hypoventilation periods. When diagnostic DLB was performed with ETT, one ED95 dose of rocuronium was administered and at the end of surgery, it was reversed with either atropine and neostigmine or sugammadex. If diagnostic DLB was performed with mask ventilation or tracheostomy, rocuronium was not used. Dexamethasone (IV 0.2 mg/kg) was administered after the induction of anesthesia to all of the patients. Rectal or IV paracetamol was used for postoperative analgesia.

All of the patients were monitored via ECG and monitoring noninvasive blood pressure, pulse oximetry, end-tidal CO_2_, and temperature during the surgery. Blood sugar was controlled at the beginning of the surgery and also when necessary.

Airway management types were recorded as endotracheal intubation (ETT), tracheostomy, and apneic anesthesia with intermittent mask ventilation. The airway management type was decided according to the child’s age and possible pathology by the ENT surgeon and anesthesiologist. All of the procedures were performed by the same ENT surgeon, who specialized in pediatric ENT (T.E.), and anesthesia management was performed by a pediatric anesthesiologist (E.O.). When the child was old enough and difficult intubation was not anticipated, such as in the case of a large laryngeal web, endotracheal intubation was the first choice for airway management. Ventilation from a tracheostomy cannula was performed when the child already had a tracheostomy or the ENT surgeon planned a tracheostomy before the operation. Apneic anesthesia with intermittent mask ventilation was the choice for airway management when an ETT would block the view of the surgeon during DLB.

Oxygenation was provided with a catheter placed from the nostril to the oropharynx when the selected airway management technique was apneic oxygenation with intermittent mask ventilation. Additionally, the 0.5 L/kg/min O_2_ flow rate was adjusted during apneic periods. When desaturation below 95% occurred or when the surgeon was changing the equipment, mask ventilation started immediately with 100% oxygen.

The descriptive statistics and frequencies were calculated by using the IBM SPSS Statistics 24 program. All statistical analyses were performed with the Statistical Package for the Social Sciences (version 22.0; SPSS Inc., Chicago, IL, USA). Results were given as percentages. The numerical variables such as age and operation time were given as mean ± SD.

## 3. Results

A total of 105 interventions were performed for 57 pediatric patients between 2018 and 2023. The ages of the patients varied between 0 and 196 months (mean ± SD; 35.77 ± 49.13). Thirty-three (56.1%) of the fifty-seven patients were younger than one year old, 28.1% of the patients were female (*n* = 16), and 71.9% were male (*n* = 41). Fifteen of the patients (26.3%) were followed-up on in the intensive care unit (ICU) and transported to the ICU after their operation.

Most of the DLBs were performed for subglottic stenosis (29.1%). A total of 17.2% of the DLBs were diagnostic DLBs without any laryngeal pathology, 11.9% were performed for recurrent papillomatosis (Figure 1), 6.8% for synechia, 6.9% for fistula, 5.9% for vocal cord mass, and 4.9% for laryngeal web (Figure 2) (Table 1).

The patients younger than 12 months old were mostly scheduled DLB for subglottic stenosis (42.5%). Although their percentage was lower, subglottic stenosis was also the most common pathology for DLB for patients older than 12 months (15.4%).

The mean operation time was 66.33 ± 38.25 min. Thirty-one (54.4%) of the patients had an intubation history before DLB and all of the patients diagnosed with subglottic stenosis had a previous intubation history and ICU stay because of either major surgery or prematurity.

Airway management was achieved with apneic anesthesia and intermittent mask ventilation (*n* = 39, 37.2%), tracheostomy cannula (*n* = 31, 29.5%), and intubation (*n* = 35, 33.3%). Intermittent mask ventilation was the selected airway management type in 66.0% of the cases in patients under one year old and 13.8% of the cases in patients older than one year (Table 2). Subglottic stenosis (31%) was the most common pathology in which intermittent mask ventilation was performed and the second pathology was diagnostic DLB (25.5%).

The complication rate related to the airway was 2% (*n* = 2). Laryngospasm was the complication in both of the cases and was treated with steroids and positive pressure ventilation via the anesthesia mask. The airway management type was apneic anesthesia and intermittent mask ventilation in both cases.

## 4. Discussion

In this study, we summarized the pediatric DLB patients’ airway management in a single university hospital. We found that although the percentage of three airway management techniques was similar when all the cases were analyzed, intermittent mask ventilation was the most common technique for infants during DLB. We also found that the most common pathology requiring DLB in pediatric patients was subglottic stenosis.

Laryngomalacia, the collapse of supraglottic tissue during inspiration [3,4], laryngeal cleft; an abnormal contact between the larynx and esophagus [5], recurrent laryngeal papillomatosis [6,7,8], subglottic cysts [6,9], subglottic hemangioma [10], subglottic stenosis, a tracheal lumen ≤4 mm for full-term neonates, and ≤3 mm premature [11] and cord vocal paralysis are the pathologies that warrant DLB, especially in early childhood. Laryngomalacia is the most common reason for airway obstruction in infants [12]. Unlike the incidences given in the literature, we found that the most common pathology causing difficult respiration resulting from the DLB procedure was subglottic stenosis in our hospital. Because it is a tertiary university hospital, complicated cases are referred to our hospital. Also, there are pediatric and premature intensive care units and complex cardiac and pediatric surgeries can be performed. These could be the reason for more complicated and rare causes among the most common pathologies in our DLB cases.

The most common benign laryngeal tumor in the pediatric age group is recurrent papillomatosis [13]. Compatible with the literature, recurrent papillomatosis was the most common benign tumor in our DLB interventions. In a review, it was reported that the mean number of surgeries for recurrent papillomatosis per child is 4.4 [14]. Although tumors are the rarest causes of respiratory distress in children, because of the need for recurrent surgeries and our hospital being a center for pediatric laryngeal pathologies, benign tumors such as papilloma, hemangioma, nodules, and cysts are the second most common pathology in our study. Malignant tumors are even rarer than benign tumors in children. There was not any patient with a malign tumor in our DLB series.

Direct laryngoscopy can be performed in older or younger age groups. Vocal cord polyps or malign tumors are the most common causes of adult DLB. A small endotracheal tube, even during operational DLB, for adult patients can provide a good view for the surgeon and adequate ventilation. The diameter of the subglottis, the narrowest part of the trachea, in newborns is 5 to 7 mm, so even intubating with a 2.0 mm intubation tube could block the surgeon’s view; therefore, tubeless anesthesia techniques are preferred during pediatric DLB [2]. In our study, we found that more than half of the patients had their DLB operation performed with a tubeless technique, either apneic anesthesia with intermittent mask ventilation or tracheostomy. Since nearly 60% of our patients were infants, this is understandable. Additionally, nearly half of our infant DLB patients were diagnosed with subglottic stenosis, where finding a suitable endotracheal tube size is even harder. We apply nasal oxygen during apneic periods to prevent desaturation for a longer time and to enable the surgeon to view the larynx properly. At this point, the experience of the surgeon is very important.

In a study involving 64 adults, a low-flow oxygenation technique via a catheter was used, similar to our practice [15]. The authors reported that the low-flow oxygenation technique is an effective and inexpensive technique for microlaryngoscopy. Oxygen delivery during the apneic period is a well-known technique for laryngeal surgery or difficult airway management [16]. The location of oxygen delivery can vary between the nose, with a nasal cannula, and trachea. Achar et al. [17] compared two locations of nasal oxygen prongs and nasopharyngeal catheters during a simulation of difficult intubations and found that the nasopharyngeal catheter is better for oxygenation. It is reported that oxygen flow anywhere in the upper airway improves apneic time but the more distal is associated with greater influence [16]. In our daily practice, we use a nasal catheter placed in the oropharynx in our pediatric DLB patients during the apnea period.

The optimal flow rates during apneic oxygenation are unknown. Riva et al. [18] showed that both low flow, 0.2 L/kg/min, and high flow, 2 L/kg/min nasal oxygen, are effective for longer apnea times. Lyons et al. [19] discussed that because pediatric patients are more prone to airway closure, high-flow oxygenation may be a better option during apnea periods. Milesi et al. [20] showed that a minimum of 2 L/kg/min oxygen flow is necessary for a pharyngeal pressure higher than 4 cm H_2_O. The patient population of this study was pediatric patients with bronchitis, so it may not reflect the physiology of the children scheduled for DLB with normal pulmonary function. We use 0.5 L/kg/min O_2_ flow rates from a nasopharyngeal catheter during the apnea periods.

One of the limitations of this study, although we observed the end-tidal CO_2_ levels of the children we ventilated from an intubation tube or tracheostomy cannula during DLB, was that we did not measure the CO_2_ levels of the patients on which we performed apneic anesthesia and intermittent mask ventilation. Cook et al. [21] showed that with 1 L/min oxygen delivery, the CO_2_ levels of children increase quickly in the first minute of apnea (1.6 kPa) and increase linearly in the following minutes (0.56 kPa/min). Also, it was found that these changes in CO_2_ levels are similar between infants, children, and adults [22]. Unlike these studies, Humphreys et al. [23] reported that there was not a significant difference between apneic patients receiving high-flow oxygen or only jaw-trust, and over a period of 5 min CO_2_ levels increased by 0.32 kPa/min. Similarly, Riva et al. [18] found that a CO_2_ increase with high-flow (THRIVE) and low-flow nasal oxygen in pediatric patients is comparable. At the end of apnea periods, we ventilated the patients with an anesthesia mask and let the surgeon continue the procedure only when the saturation was 100% and the ETCO_2_ was between normal values. The second limitation is that we did not measure the apneic times of the patients for each apnea time point. Patel et al. [24] found apnea time to 90% saturation was 119 s in infants and 160 s in children between 2 and 5 years old. Recent studies showed that oxygen insufflation prolongs the apnea time in pediatric patients [25,26]. This study was planned as a retrospective study, and unfortunately, the apnea times were not recorded in the anesthesia charts. A prospective study could be planned to measure the apnea times and CO_2_ values for different age groups by using this method during DLB.

## 5. Conclusions

In conclusion, we reported our experiences during pediatric DLB surgeries. Airway management can change according to the age of the child and existing pathology when performing DLB procedures. Especially for infant patients, tubeless techniques are more convenient for the surgeon to view the larynx and perform the required intervention to provide an open airway and ease the child’s respiration. Endotracheal intubation and tracheostomy are the safest airway management types during oropharyngeal surgeries. Our results showed that intermittent mask ventilation with a nasopharyngeal oxygen cannula and 0.5 L/kg/min oxygen can be a safe airway management technique for infant DLBs. However, the anesthesia team should discuss the laryngeal pathology, type of operation, and ease of intubation with the surgeon and learn the possible difficulties related to the airway before the surgery so they can prevent or be ready for inadvertent events. We recommend that the anesthesiologist should see the indirect endoscopic view of the larynx, if available, before planning the airway management.

## Figures and Tables

**Figure 1 medicina-60-01729-f001:**
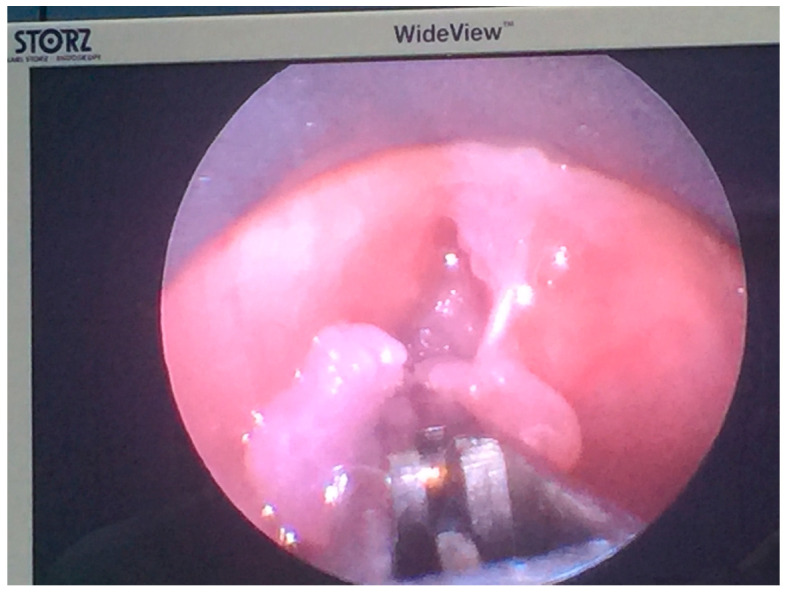
Resection of recurrent laryngeal papillomatosis in a 6-year-old girl.

**Figure 2 medicina-60-01729-f002:**
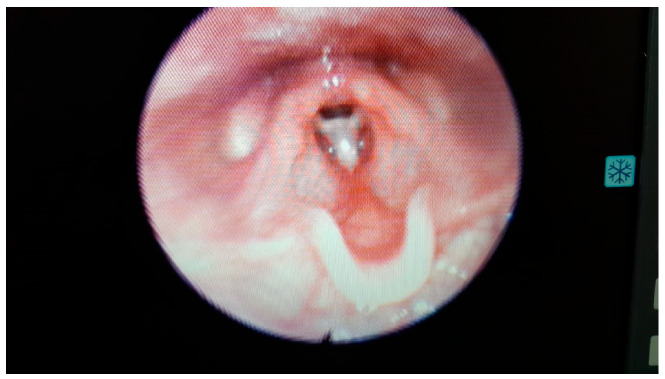
Laryngeal web in a 26-month-old baby.

**Table 1 medicina-60-01729-t001:** The pathologies of the patients that required direct laryngoscopy (DLB: direct microlaryngobroncoscopy).

Pathology	Percent
Subglottic stenosis	29.1%
Diagnostic DLB	17.2%
Recurrent papillomatosis	11.9%
Glottic or subglottic synechia	6.8%
Fistula	6.9%
Vocal cord mass	5.9%
Laryngeal web	4.9%
Subglottic hemangioma	4.8%
Abscess	3.9%
Tracheomalacia	2.9%
Foreign body	2%
Laryngomalacia	1%
Vallecula cysts	1%
Epiglothopexy	1%

**Table 2 medicina-60-01729-t002:** Airway management type and relation with age (ETT: endotracheal tube).

		Airway Management		
		ETT	Tracheostomy	Mask
Age	<1 year	6 (12.8%)	10 (21.3%)	31 (66.0%)
	>1 year	29 (50.0%)	21 (36.2%)	8 (13.8%)

## Data Availability

The datasets generated during and/or analyzed during the current study are available from the corresponding author upon reasonable request.

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
