# Peer review of "Pediatric Microlaryngoscopy Experiences in a Tertiary Hospital: A Retrospective Analyses of 105 Procedures"

_medicina, 2024, doi:10.3390/medicina60111729_

Round 1
Reviewer 1 Report
Comments and Suggestions for Authors
I read the manuscript with the interest .
The manuscript presents the retrospective observational cohort study of the pediatric microlaryngoscopy experience in a single tertiary centre.
The aim of the study was to investigate the paediatric DLB patients performed in ear, nose, throat (ENT) operation room for airway management type and complications. This is highly specialized procedure performed in highly specialized tertiary hospital. For that reason, the manuscript is very relevant for the field since there are no many centres with such high performance. As the result of this high specializations and high number of procedures, the staff is obviously well skilled and have very low rate of complications.
I thank the authors for the presentation of the manuscript and their initiative to share their experience. However, I suggest the authors to improve their manuscript, particular the method part, in order to be interesting enough for publishing.
The abstract is understandable to read in isolation.
The background and relevant research are briefly presented in the introduction, and the authors’ aim(s) are pointed out.
The methods are presented but too general to be reproducible. To be consistent with the aims, the authors should be focused on the airway management methodology. Most relevant details are missing, particularly details about airway equipment, oxygen devices, respiratory equipment, monitoring equipment, anaesthesia drugs and anaesthesia techniques. In addition more details about airway operator/performer should be presented ( e.g.specialist or resident, ENT or anaesthesiologist, paediatric anaesthesiologist) . These data could be further analysed and give us the insight in the authors’ airway management techniques which could be as a good clinical example adopted in clinical practice of other institutions.
The results come from the descriptive statistics. However, these results tell very little. As there are very few data about the patients, the relevant analytic statistics is missing.
The discussion is well written. The results are compared with the literature.
The conclusions are partially consistent with the evidence and arguments presented. ( Line 186-194). It is not clear what about the team should discuss before the procedure. From the presented results it seems that there is no problem with the airway management for these procedures.
The relevant ethical statements are included.
Comments on the Quality of English Language
Minor editing of English language required.
Author Response
Thank you for your valuable comments about our paper.
Comment: The methods are presented but too general to be reproducible. To be consistent with the aims, the authors should be focused on the airway management methodology. Most relevant details are missing, particularly details about airway equipment, oxygen devices, respiratory equipment, monitoring equipment, anaesthesia drugs and anaesthesia techniques. In addition more details about airway operator/performer should be presented ( e.g.specialist or resident, ENT or anaesthesiologist, paediatric anaesthesiologist) . These data could be further analysed and give us the insight in the authors’ airway management techniques which could be as a good clinical example adopted in clinical practice of other institutions.
Response: The methods part is extended. They are highlighted in methods section.
Comment: The results come from the descriptive statistics. However, these results tell very little. As there are very few data about the patients, the relevant analytic statistics is missing.
Response: This is a retrospective cross-sectional study performed by the same anesthetist so unfortunately an analytic statistics couldn't performed. We reported our experiences during pediatric DLBs and define the airway management techniques we use.
Comment: The conclusions are partially consistent with the evidence and arguments presented. ( Line 186-194). It is not clear what about the team should discuss before the procedure. From the presented results it seems that there is no problem with the airway management for these procedures.
Response: The conclusion part is enlarged.
Reviewer 2 Report
Comments and Suggestions for Authors
This observational qualitative study is an interesting one but but has been dealt in a very superficial manner. Just the number of cases and types of procedures have been elaborated and not the details of how the procedures were performed. It could be emphasised.
Other points include:
1. Anaesthetist: To call the specialty anaesthesiologists pl
2. Line 65-66, 81, 101: There is a symbol in the lines which is not getting copied here.meanSD; 65 35.7749.13
3. Line 88-90: The most common pathology intermittent mask ventilation performed was subglottic stenosis (31%) and second pathology was diagnostic DLB. The gramer may be modified
4. References 15-17 are added but no such analysis was done bt the authors
5. One reference is from 1997. Can be replaced
Author Response
Thank you for your valuable comments that would improve our paper.
Comment: This observational qualitative study is an interesting one but but has been dealt in a very superficial manner. Just the number of cases and types of procedures have been elaborated and not the details of how the procedures were performed. It could be emphasised.
Response: The methods section is enlarged.
Comment: Anaesthetist: To call the specialty anaesthesiologists pl
Response: It is corrected.
Comment: Line 65-66, 81, 101: There is a symbol in the lines which is not getting copied here.meanSD; 65 35.7749.13
Response: It is corrected.
Comment: Line 88-90: The most common pathology intermittent mask ventilation performed was subglottic stenosis (31%) and second pathology was diagnostic DLB. The gramer may be modified
Response: It is modified.
Comment: References 15-17 are added but no such analysis was done bt the authors
Response: Those literatures were added yo discuss our apneic oxygenation technique.
Comment: One reference is from 1997. Can be replaced
Response: Mostly we used the latest literature but we believe that to cite the paper that the related data was first published is more valuable as you would also appreciate.
Round 2
Reviewer 1 Report
Comments and Suggestions for Authors
The authors did make an effort in improvement. However, the methods are not given in details enough to be reproducible and the conclusions are not supported by the evidence from the study.
Comments on the Quality of English LanguageMinor editing of English language required.
Reviewer 2 Report
Comments and Suggestions for Authors
There has not been a substantial change in the study design which perhaps is not possible at this stage.